

# HybridFormer: a convolutional neural network-Transformer architecture for low dose computed tomography image denoising

Shanaz Sharmin Jui*, Zhitao Guo*, Rending Jiang, Jiale Liu and Bohua Li

School of Electronics and Information Engineering, Hebei University of Technology, Tianjin, Beichen, China
* These authors contributed equally to this work.

## ABSTRACT

Low-dose computed tomography (CT) is a potent strategy to minimize X-ray radiation and its detrimental effects on patients. However, reducing radiation significantly boosts noise in reconstructed images, causing blur and obscuring critical tissue details. This obscurity poses significant challenges for doctors in making accurate diagnoses. Traditional techniques like sinogram domain filtration and iterative reconstruction algorithms require inaccessible raw data. Thus, this article introduces HybridFormer, a revolutionary image-denoising model utilizing the Residual Convolution-Swin Transformer Network, designed to enhance images while preserving vital details. Firstly, this algorithm constructs residual convolution for local feature extraction and Swin Transformer for global feature extraction, boosting denoising efficacy. Secondly, to address texture detail errors, we introduced a combined attention transformer unit (CATU) with a cross-channel attentive fusion layer (CCAFL), integrated with residual blocks to form a residual convolution and Swin Transformer Fusion Block (RSTB). Finally, using RSTB, we developed a deep feature refinement module (DFRM) to preserve image details. To avoid smoothing, we combined multi-scale perceptual loss from ResNet-50 with Charbonnier loss into a composite loss function. Validated on the AAPM2016 Mayo dataset, HybridFormer outperformed other state-of-the-art algorithms, achieving improvements of 0.02 dB, 0.16%, and 0.28% in PSNR, SSIM, and FSIM, respectively. Compared with other advanced algorithms, the proposed algorithm achieved the best performance indicators, confirming its superiority.

## INTRODUCTION

X-ray computed tomography (CT) has found extensive applications in clinical, industrial, and diverse fields. The widespread application of medical CT has heightened concerns regarding the accumulated radiation dose received by patients. To address this challenge, researchers and medical professionals are actively exploring various innovative approaches, including ongoing research endeavors dedicated to investigating novel

Corresponding author
Zhitao Guo, mrnow@hebut.edu.cn

methods and refining existing techniques to minimize CT radiation dose while maintaining image quality (*Huynh et al., 2016*; *Lell et al., 2015*; *Brenner & Hall, 2007*; *Coxson et al., 1999*). The ultimate objective is to strike a balance between providing essential diagnostic information to patients and minimizing their radiation exposure. There have been several studies on image denoising using deep neural networks. Multiple studies have proposed various deep-learning architectures for image denoising. *Zhang et al. (2017)* introduced a deep convolutional neural network (CNN) with skip connections. *Hsieh (1998)* focused on radon domain adaptive filtering for low-dose CT images. *Kim, Lee & Lee (2016)* proposed a residual encoder-decoder CNN (REDCNN). *Zhang et al. (2018)* utilized a multi-scale deep CNN (MSDCNN). *Karimi & Ward (2016)* developed a clustering-based CT denoising algorithm. *Tai, Yang & Liu (2017)* added a spatial attention module (SAM) to their CNN. *Xie et al. (2017)* used a multiplier alternating direction algorithm for CT imaging. *Zhang, Zuo & Zhang (2019)* proposed a residual dense network. *Yang et al. (2019)* combined a generic deep denoiser with a detail enhancement module. *Wang et al. (2022)* employed channel and spatial attention with multi-scale residual fusion. *Zhang, Li & Li (2018)* used a dual-attention network. *Zhang et al. (2021)* hybridized deep CNN and Swin Transformer. *Liang et al. (2021)* introduced SwinIR, a Swin Transformer-based network. *Lu et al. (2001)* applied a distributed logarithmic transformation and Wiener filtering. *Zhang et al. (2016)* designed a Gaussian mixture Markov random field model (GM-MRF). *Liu et al. (2015)* used total variation (TV) regularization with median prior constraints. *Chen et al. (2009)* integrated adaptive weighted non-local means. *Xu et al. (2012)* and *Bai et al. (2013)* utilized dictionary representation for detail preservation. *Chen et al. (2016)* combined block matching and 3D filtering with context optimization. *Zamyatin et al. (2014)* improved multiscale filtering and TV cost functions. *Hashemi et al. (2014)* designed a non-local total variation denoising method. *Chen et al. (2013)* proposed a post-processing algorithm based on dictionary learning. Deep neural networks (DNNs), *Yang et al. (2017)*, *He et al. (2016)*, *Simonyan & Zisserman (2014)*, *Goodfellow et al. (2020)* including CNNs and generative adversarial networks (GANs), have significantly advanced image denoising techniques.

DNNs, including CNNs and GANs, have revolutionized image denoising (*Zhang et al., 2017*; *Mao, Shen & Yang, 2016*), achieving state-of-the-art results. CNNs capture spatial information, and recent advances like ResNets and attention mechanisms have improved performance. GANs can produce visually appealing results but are unstable and difficult to train (*Ledig et al., 2017*). *Wang et al. (2021)* proposed DuDoTrans, a Transformer-based model for CT image reconstruction, overcoming CNN limitations. Sparse representation and dictionary learning methods (*Elad & Aharon, 2006*; *Mairal, Elad & Sapiro, 2008*) use sparsity to remove noise while preserving image structures. Some studies combine CNNs with analytical or iterative reconstruction for better low-dose CT images (*Wu et al., 2017*; *Kang et al., 2018*). Non-local means filtering (*Buades, Coll & Morel, 2005*; *Rodrigues Jonathan et al., 2019*; *Yu et al., 2021*) leverages self-similarity to reduce noise while preserving edges and textures (*Fan, Liu & Liu, 2022*). *Liang et al. (2020)* designed a densely connected denoising model for low-dose CT images, while *Chen et al. (2022)* revealed that

non-linear activation functions may not be necessary for image restoration. Recent breakthroughs integrate Transformer architectures for global context modeling: TransCT (*Kang et al., 2018*) and Uformer (*Wang et al., 2022*) synergize self-attention mechanisms with CNNs, while dual-branch deformable Transformers (*Gholizadeh-Ansari, Alirezaie & Babyn, 2020*) enhance spatial adaptability. Cutting-edge solutions like Restomer (*Zamir et al., 2022*) address computational complexity through efficient tokenization, and WavResNet (*Kang et al., 2018*) simultaneously optimizes sinogram and image domains. These related works provide a comprehensive overview of the different approaches (*Niu et al., 2024*) used for image denoising. Each approach has its (*Zhang, Zhang & Lu, 2010*) strengths and weaknesses, and the choice of method depends on the specific traits of the noise and the desired output.

This article introduces an innovative network architecture named Hybrid Former, specifically designed for preserving crucial details in images. Hybrid Former harmoniously integrates the strengths of residual convolution for capturing local features and Swin Transformer for global feature extraction, yielding remarkable results in de-noising tasks. Furthermore, an internal feature extraction module is meticulously crafted and seamlessly integrated into the encoder framework to elevate the model's proficiency in extracting vital image details. During the network's training phase, a sophisticated composite loss function, amalgamating Charbonnier loss and multiscale perceptual loss, is devised to mitigate the risk of over-smoothing image edges that may arise from using a solitary loss function. Ultimately, through a comprehensive series of ablation and comparative experiments, Hybrid Former's superiority in denoising low-dose CT images is unequivocally demonstrated through rigorous numerical evaluations of various metrics.

## METHODOLOGY

### Noise reduction model

Because of the significant noise in low dose computer tomography (LDCT) images, they are frequently perceived as possessing substandard quality. This article introduces an advanced algorithm leveraging deep learning techniques to eliminate noise from these images and elevate their quality. Mathematically, the challenge of denoising LDCT images can be formulated as follows:

Let $X \in R^{m \times n}$ represent an LDCT image and $Y \in R^{m \times n}$ represent a normal-dose CT image.

$$X = \sigma(Y). \tag{1}$$

Equation (1), $\sigma$ represents the process by which noise degrades the image quality. Reducing noise from LDCT images to produce NDCT images can be conceptualized as the inverse of this degradation process. The ultimate objective is to discover a function that can accurately map LDCT images to their corresponding NDCT images while minimizing a specified cost function.

$$\arg \min_{F} ||F(X) - Y||_2^2 \tag{2}$$

where $F$ is the strongest approximation of $\sigma^{-1}$ and represents a deep-learning neural network based on a learnable transformer architecture. The detailed analysis process of the denoising method is shown in the following subsections.

## Overall network design

As shown in Fig. 1, the proposed model is a U-shaped hierarchical network with skip connections between the encoder and decoder. For an LDCT image $X \in R^{C \times H \times W}$ input into the network, a $3 \times 3$ convolutional layer is first used to extract low-level features, which are then input into the encoder stage. In the encoder, each stage contains several RSTBs and a down-sampling layer. Simultaneously, several DFRMs run in parallel in each encoder stage. The RSTB and DFRM structures are illustrated in Figs. 1A and 1B, respectively. Each RSTB includes a Combined Attention Transformer module and a residual convolutional module, fusing these two modules through two $1 \times 1$ convolutional layers, splitting, concatenation, and residual connections. In the down-sampling section, a $4 \times 4$ convolutional layer with a stride of 2 is used to down-sample the feature map, halving the output feature size and doubling the number of channels. Subsequently, a bottleneck layer with an RSTB is added at the end of the encoder. At this stage, the RSTB can capture longer dependencies.

The feature reconstruction stage consists of a decoder with the same number of stages as the encoder. Each stage includes an up-sampling layer and several RSTBs identical to those in the encoder. In the up-sampling section, a $2 \times 2$ deconvolutional layer with a stride of 2 is selected for up-sampling. The size of the up-sampled features is doubled, while the number of channels is halved. The features input into the RSTBs come from the encoder's output through skip connections and the up-sampled features. After several decoder stages, a $3 \times 3$ convolutional layer is applied to generate the residual image R. Finally, the LDCT image is added to the residual image to obtain the restored image Y, *i.e.*, Y = X + R.

## Feature extraction and global feature extraction layer

### Residual convolution and swin transformer fusion module

The architecture of the RSTB, illustrated in Fig. 1A, involves splitting the input feature X into two halves, $X_1$ and $X_2$, *via* a $1 \times 1$ convolutional layer and a splitting mechanism. This segmentation is mathematically expressed as:

$$X_1, X_2 = Split(Conv(X)). \tag{3}$$

Subsequently, $X_1$ and $X_2$ are processed through a residual convolutional block and a Combined attention Transformer (CATU) module, respectively, producing:

$$Y_1, Y_2 = ResConv(X_1), CATU(X_2). \tag{4}$$

ResConv signifies a residual convolutional block, detailed in Fig. 1C, comprising convolutional layers with activation functions and residual connections. The CATU module, introduced later, is a combined attention transformer.

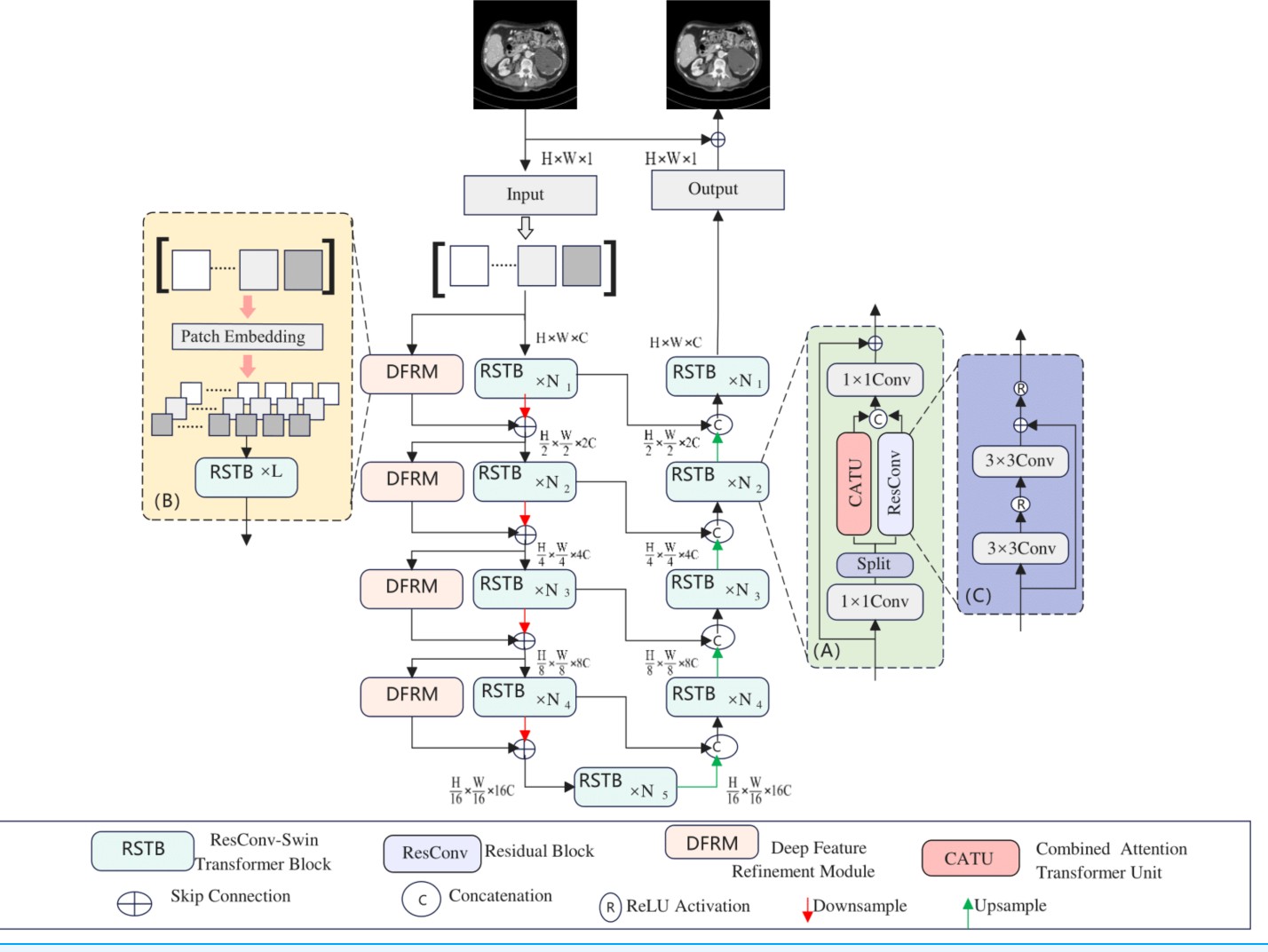

**Figure 1 Hybridformer structure diagram.**

Afterward, $Y_1$ and $Y_2$ are concatenated and passed through another $1 \times 1$ convolutional layer, with a residual connection to X, yielding the final RSTB output:

$$Z = Conv(Concat(Y_1, Y_2)) + X. \qquad (5)$$

The RSTB harmoniously combines the local processing capabilities of residual convolutions with the non-local modeling strengths of the Swin Transformer. The $1 \times 1$ convolutional layers blend information from the residual convolutional block and CATU. Furthermore, the split and concatenation operations optimize computational efficiency and parameter count.

## Deep feature refinement module

This article introduces the deep feature refinement module (DFRM), integrated into the encoder, to bolster the model's extraction of intricate details from input features. The

DFRM's structure is depicted in Fig. 1B. Initially, it performs patch embedding on input features, halving their size. These internal feature blocks are then processed through multiple RSTBs. In scenarios with small feature blocks, the DFRM excels at capturing fine details.

Operating in parallel with the main encoder block, each DFRM enhances the encoder's output. Except for the initial stage, DFRM features are summed with encoder features and fed into the subsequent model stage. Consequently, each encoder stage can be formulated as:

$$Z_i = RSTB_i(X_i) + RSTB_i^{'}(P(X_i)) \tag{6}$$

where $i$ denotes the encoder stage, $X_i$ represents input features, $RSTB_i$ signifies the main encoder block, $RSTB_i^{'}$ indicates the parallel $RSTB_i$ within the DFRM, and $P$ represents the internal feature map creation process.

## Combined attention transformer unit

In sophisticated Transformer architectures designed for image feature extraction, the conventional approach involves computing global self-attention across the entire image. However, this method entails a substantial computational burden that grows quadratically with the input image's size, leading to considerable processing overhead. To mitigate this issue, the Swin Transformer introduces innovative techniques such as window attention (W-MSA) and shifted window attention (SW-MSA), which build upon the traditional Transformer framework. The Swin Transformer achieves significant computational efficiency by alternately applying W-MSA and SW-MSA across adjacent Transformer layers. Nevertheless, the restricted information scope of window attention mechanisms may give rise to inaccuracies in texture feature restoration when deploying the Swin Transformer in denoising networks. This problem could be alleviated if the model could access a broader range of input data.

Regarding channel attention, it leverages global information to assign weights to each feature channel, activating more pixels within the input features. Integrating a channel attention module into the Swin Transformer architecture enhances the network's ability to express features. Traditional channel attention modules consist of stacked convolutional layers followed by an attention layer. Still, the attention layer typically only receives the output of the final convolutional layer, leading to a disconnect between the attention weights and earlier convolutional layer outputs. In response, this article introduces the Cross-Channel Attentive Fusion Layer (CCAFL).

CCAFL begins by processing the input features through two 3 × 3 convolutional layers with a stride of 1 and an activation function. To optimize computational costs, the first convolutional layer reduces the number of output feature channels to one-third of the original, which is then restored to the original count by the second convolutional layer. Both the activation function's output and the second convolutional layer's output are fed into the CCAFL independently. Within CCAFL, the two input features undergo global

average pooling to extract global information, which is then processed through respective convolutional and batch normalization layers. The combined results are passed through a ReLU activation function for non-linear transformation, followed by another convolutional layer and a sigmoid function for non-linear mapping. Finally, this processed result is multiplied by the output of the last convolutional layer to produce a channel feature vector refined by the attention mechanism.

In the Combined Attention Transformer Unit (CATU), the input features are first normalized and then processed using (S)W-MSA to capture local information. Simultaneously, global information is introduced through CCAFL to compute channel attention weights, which weigh the features using global information, thereby enhancing the utilization of input information. To prevent optimization conflicts between CCAFL and (S)W-MSA, the CCAFL's output is scaled by a small constant α before being added to the (S)W-MSA and residual connections. The comprehensive computation process of CATU entails these steps:

$$\begin{cases} X_{LN} = LN(X_{in}) \\ X_1 = MSA(X_{in}) + \alpha CCAFL(X_{LN}) + X_{in} \\ X_{out} = X_1 + MLP(LN(X_1)) \end{cases} \tag{7}$$

$X_{LN}$ and $X_1$ denote intermediate representations, with $LN$ signifying layer normalization. $X_{out}$ stands for the output generated by CATU, while $MLP$ refers to a sophisticated multilayer perceptron.

In the (S)W-MSA module, the input feature map $X_{LN} \in R^{C \times H \times W}$ is divided into $HW/M^2$ non-overlapping local windows, each size $M \times M$, and self-attention is computed within each window. For the local window feature $X_W \in R^{M^3 \times C}$, the $Q$, $K$, and $V$ matrices are computed as follows:

$$\begin{cases} Q = X_W \times P_Q \\ K = X_W \times P_K \\ V = X_W \times P_V \end{cases} \tag{8}$$

where $P_Q$, $P_K$, and $P_V$ are projection matrices, and $Q$, $K$, $V \in R^{M^3 \times C}$, the self-attention mechanism calculation is as shown in the formula:

$$Attention = SoftMax\left(\frac{QK^T}{\sqrt{d}} + B\right)V. \tag{9}$$

In this context, $d$ signifies the dimensionality of $Q/K$, while $B$ denotes the learnable relative positional encoding. Furthermore, to facilitate efficient interaction among adjacent non-overlapping windows, W-MSA and SW-MSA are employed in an alternating sequence, with the stride size configured to be half the window size.

## Loss function

During the optimization process, various loss functions were tested to enhance the proposed network's performance (*Zhao et al., 2022*). Initially, the mean squared error (MSE) loss was used, but it led to overly smooth and blurred images. To address this, the

Charbonnier loss was employed, which measures the difference between the model's output and the normal-dose image. Additionally, a multi-scale perceptual loss function was integrated with the Charbonnier loss, using ResNet-50 as the feature extractor, to mitigate over-smoothing and blurring. The composite loss function, combining both losses, was derived by tuning the hyperparameter $\lambda$ to adjust the weight of the multi-scale perceptual loss.

The Charbonnier loss measures the difference between the model's output and the NDCT image and is defined as:

$$L_{char} = \sqrt{||F(X_i) - Y|| + \varepsilon^2} \tag{10}$$

$X_i$ represents the input image, $F$ denotes the denoising process, $Y$ is the normal-dose image, and $\varepsilon$ is a tiny constant, typically $10^{-3}$ (0.001).

ResNet-50 is selected due to its exceptional capacity to derive intricate, deep-level information from images and incorporate residual learning frameworks, thereby guaranteeing a more stable and reliable acquisition of image features during the computation of perceptual loss. The precise methodology entails stripping away the pooling and fully connected layers from ResNet-50, retaining solely the convolutional layers for further utilization, at the front of the model. When calculating the perceptual loss, the restored image and the NDCT image were input into the feature extractor for forward propagation. Then, the Charbonnier loss was computed using the features from four stages, and these values were averaged to obtain the multi-scale perceptual loss, which is expressed as follows:

$$L_{per} = \frac{1}{M} \sum_{j=1}^{M} \sqrt{||\Phi_j(F(X_i)) - \Phi_j(Y)|| + \varepsilon^2} \tag{11}$$

$X_i$ is the input image, $F$ represents the denoising process, $Y$ is the normal-dose image, and $\Phi$ denotes the pre-trained ResNet-50 model with fixed weights.

Combining the Charbonnier loss with the multi-scale perceptual loss yields the composite loss function, as shown below:

$$L_{compound} = L_{char} + \lambda L_{per} \tag{12}$$

where the weight of the multi-scale perceptual loss in the composite loss is adjusted by tuning the hyperparameter $\lambda$.

# EXPERIMENTAL RESULTS AND ANALYSIS

## Materials and network training

### Dataset

In the experimental design and results section, the dataset comprises low-dose CT scans from the "2016 NIH-AAPM-Mayo Clinic Low-Dose CT Grand Challenge" licensed by Mayo Clinic. Specifically, CT scan images with a thickness of 3 millimeters are selected, including both normal-dose and quarter-dose CT images. The training set consists of 2,039 image pairs (*Zamir et al., 2021*) from eight patients, the validation set contains 128 image pairs from one patient, and the test set utilizes 211 image pairs from another patient. Data

**Table 1 Experimental hyperparameter settings.**

| Parameter name | Parameter value |
|---|---|
| Batch size | 12 |
| Learning rate | $2 \times 10^{-4}$ |
| Optimizer | AdamW |
| Optimizer betas | 0.9 |
| Training epoch | 100 |
| Image size | $64 \times 64$ |

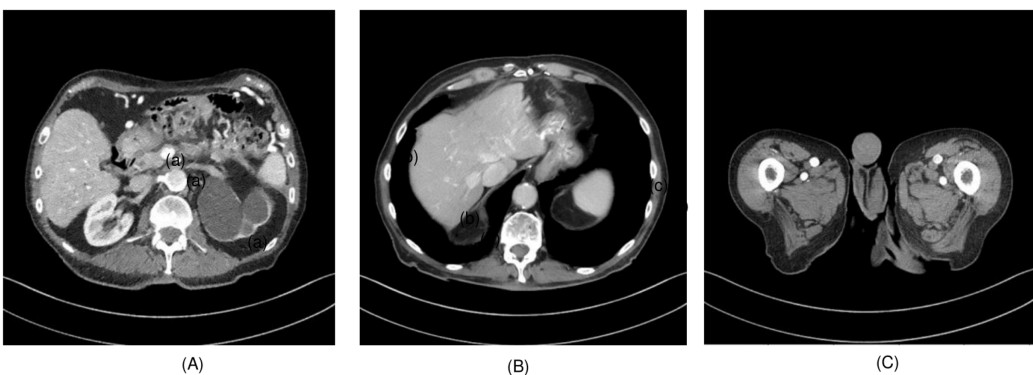

| (A) | (B) | (C) |

**Figure 2 (A) Sample 1 and (B) sample 2, (C) sample 3.**

augmentation is performed before training, involving converting the image data into PyTorch-compatible Tensor format and cropping random patches of size $64 \times 64$ from each slice. During testing, full $512 \times 512$ image slices from patient scans are directly used.

### Experimental setup

The experiments were conducted on a Windows 11 system with the PyTorch deep learning framework and CUDA-Toolkit10.1 for GPU acceleration. The hardware configuration consisted of an Intel Core i7-9700K CPU @ 3.2 GHz, 16 GB of RAM, and an NVIDIA GeForce RTX 3080 series GPU equipped with 8 GB of video memory (VRAM). During the optimization phase, we employed the highly efficient AdamW optimizer (*Kang et al., 2018*; *Wright et al., 2024*) with its default settings. We meticulously adjusted the learning rate to $2 \times 10^{-4}$ and incorporated the optimizer betas of 0.9. We implemented a stringent 100-epoch training protocol to ensure reliable model convergence. Table 1 presents the detailed hyperparameter configurations for all experiments detailed in this article, highlighting the high-performance computing resources utilized for the experiments.

### Results and analysis

To evaluate the effectiveness of the proposed algorithm, two highly detailed abdominal CT images, depicted in Fig. 2, (a) sample 1 and (b) Sample 2, were chosen as benchmarks for comparison. To perform cross-validation with various splits, sample 3 is selected, and Fig. 3 presents the comprehensive comparison of the test results for sample 3 along with

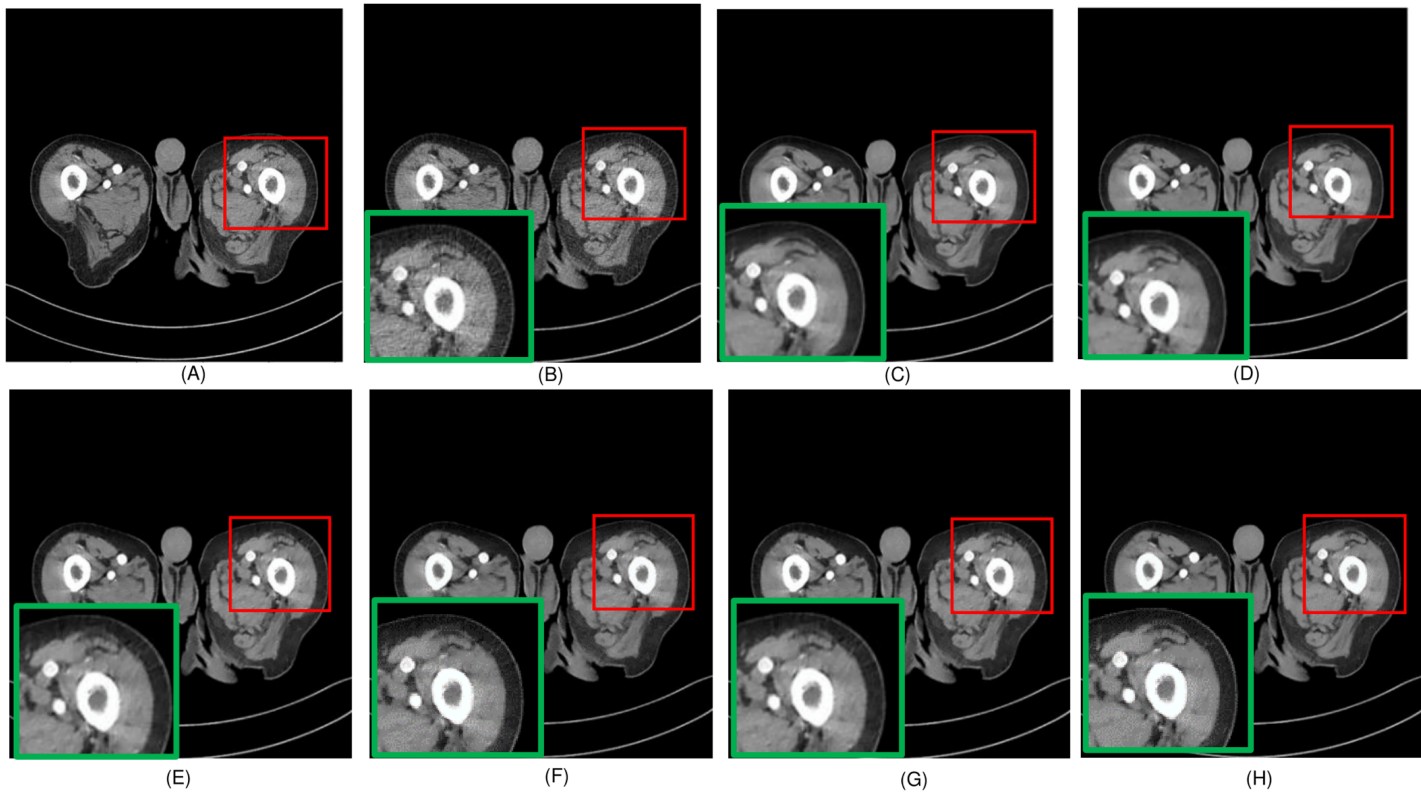

**Figure 3** Overall comparison of test sample 3 (A) NDCT (B) LDCT (C) Uformer (D) CTformer (E) Restormer (F) EDCNN (G) DDT (H) Proposed.

the magnified ROI area. We have carefully chosen these samples at random from a diverse array of patients.

In this experiment, we compared five algorithms: EDCNN (*Liang et al., 2020*), CTformer (*Dayang et al., 2023*), Restormer (*Zamir et al., 2022*), Uformer (*Wang et al., 2022*), and DDT (*Liu et al., 2023*). Our comparison also includes low-dose CT images, normal-dose CT images, and Hybrid Former. All parameters for the algorithms were set according to their respective papers. We selected a test sample containing complex image details from the test set to perform denoising processing. The upper middle side of test sample 1, containing an obvious lesion tissue area, is selected as the region of interest (ROI) for our comparison. Here, Fig. 4 displays the overall comparison of test sample 1. We magnified and displayed the comparison results along with Fig. 4. A test sample with similar background structures is selected from the test set as test sample 2 and test sample 3. Figure 5 shows the comparison results of the residual images after denoising sample 1. It can be noted that in low-dose CT images with similar background structures, the proposed method successfully eliminates noise, significantly improving image quality. Figure 6 shows the comparison results of selecting a lesion area from test sample 2 along with the ROI area. Upon close examination, it is evident that the transformer-based algorithms such as Restormer, CTformer, and Uformer are significantly similar to the proposed method. To further elucidate the superiority of the proposed method, we have

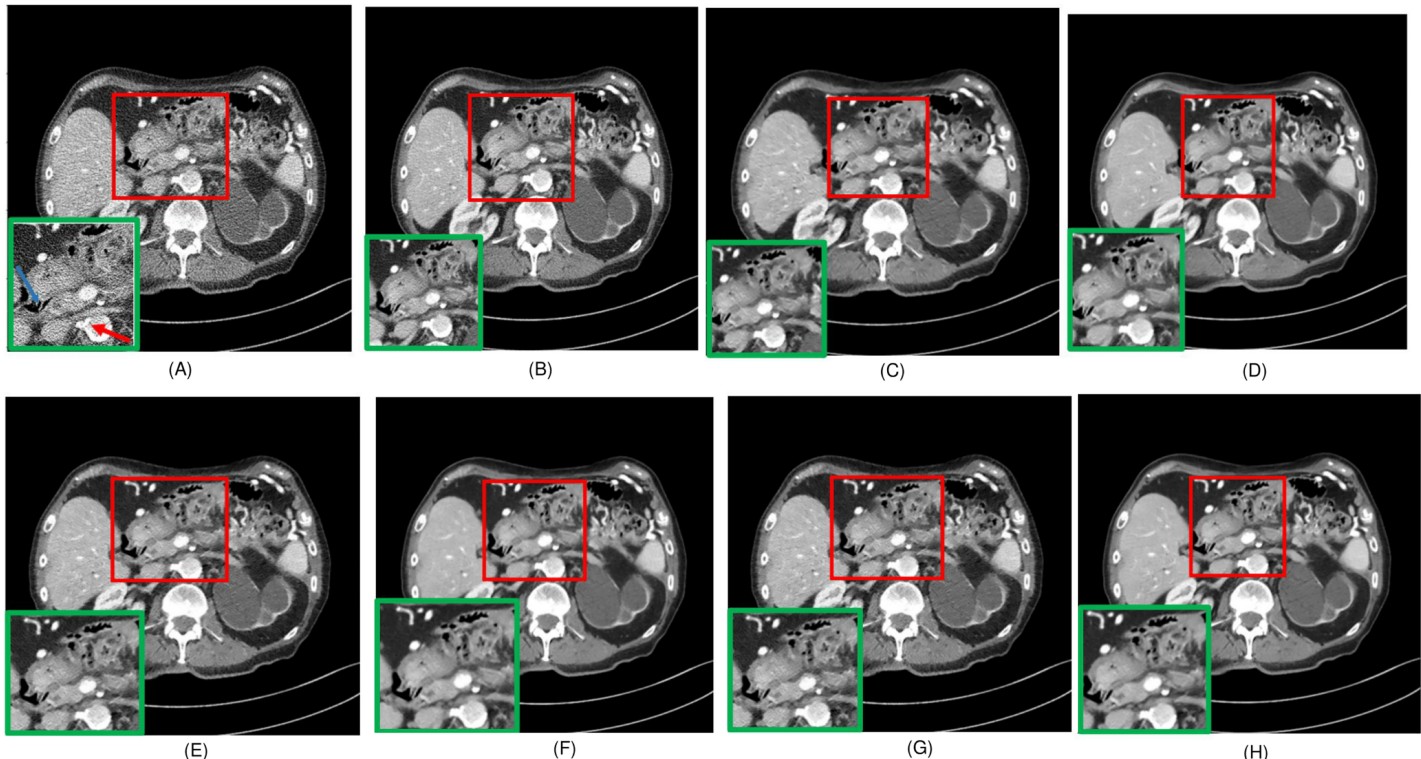

**Figure 4 Overall comparison of test sample 1 (A) LDCT (B) NDCT (C) Uformer (D) Restormer (E) EDCNN (F) CTformer (G) DDT (H) Proposed.**

included Figs. 5, 7, and 8, which depict the residual images and error map, respectively. It can be seen that the proposed algorithm can effectively remove noise from low-dose CT images with similar background structures and preserve detailed information well.

Upon examining the magnified ROI depicted in Figs. 3 and 4, it is evident that the LDCT image, plagued by substantial quantum noise, exhibits excessively blurred edge details, as highlighted by the red regions, resulting in the loss of some sharp edge features. Conversely, HybridFormer demonstrates a superior visual impact in terms of intuitive noise contrast, aligning more closely with the NDCT image than alternative approaches, as illustrated in Fig. 4. A thorough comparison of the magnified ROI among various denoising methods in Fig. 4 reveals that the proposed algorithm effectively preserves image edges. By scrutinizing the areas indicated by both red and green areas, it becomes apparent that HybridFormer offers sharper edge details and a more distinct contrast against the surrounding background. Figure 6 presents a comprehensive comparison of another exemplary specimen within test sample 2, along with an enlarged region of interest (ROI) featured, while Fig. 7 delves into a detailed comparison of residual images featured by Fig. 6. Upon visually inspecting Fig. 6, it becomes evident that the proposed method exhibits an exceptional denoising capability on CT images comprising diverse structural elements. When scrutinizing the magnified ROIs depicted in Fig. 6, it is apparent that the introduced denoising algorithms similarly demonstrate remarkable denoising proficiency

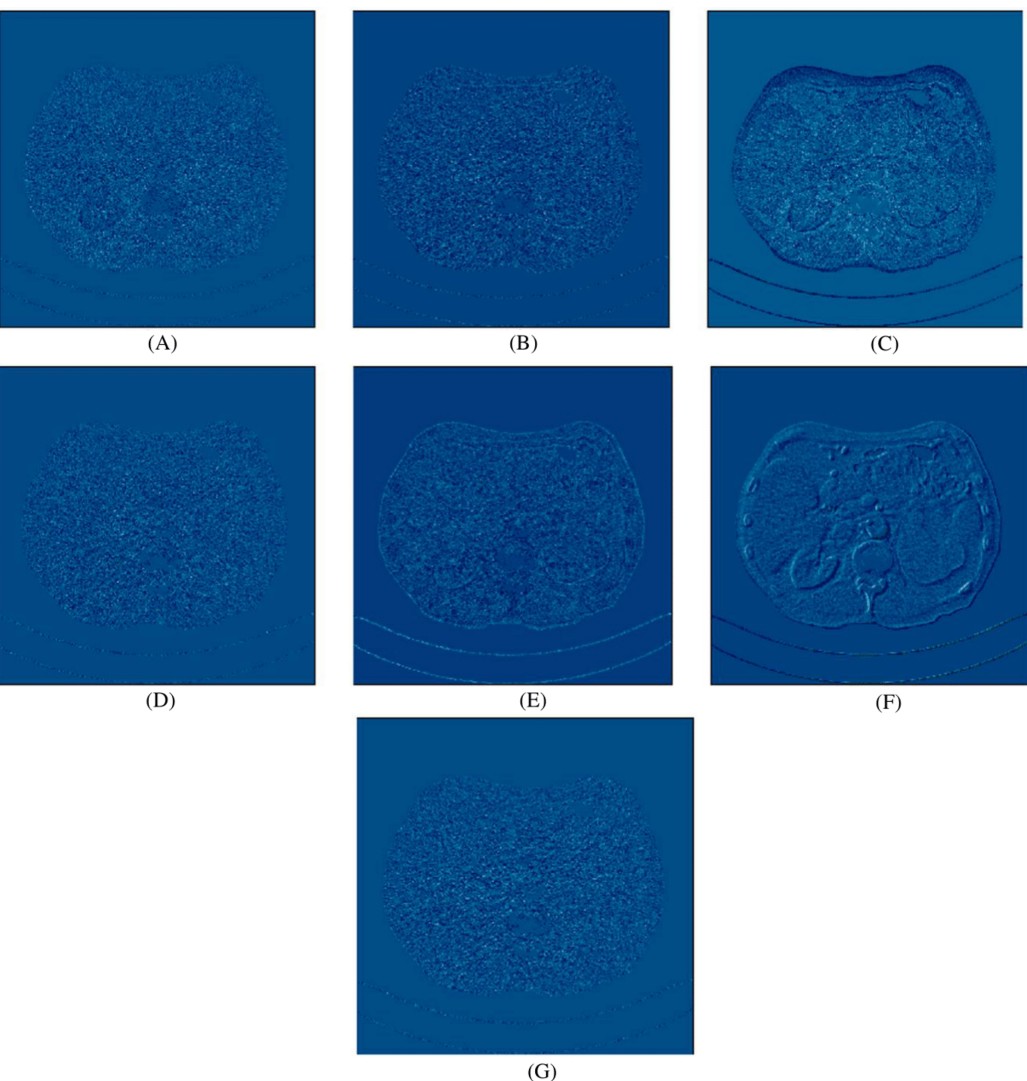

**Figure 5 Noise distribution image of sample 1 after de-noising, LDCT, and different algorithms (A) LDCT; (B) Restormer; (C) Performer; (D) CTformer; (E) EDCNN; (F) DDT; (G) Proposed.**

in areas with comparable background structures, efficiently preserving the edge contours of similar components. Notably, this algorithm not only preserves a higher degree of edge details but also attains a denoising effect that most closely approximates the quality of an NDCT image, rendering the overall image significantly cleaner.

Tables 2 and 3 display the evaluation outcomes for test sample 1, test sample 2, and test sample 3, highlighting the optimal values in each metric in bold. Meanwhile, Table 4 presents a comparative analysis of the Hybrid Former alongside other image-denoising algorithms, focusing on their overall impact across the entire test dataset. The findings underscore notable advancements in performance metrics compared to existing models. In essence, the Hybrid Former excels in achieving superior denoising results across the entire

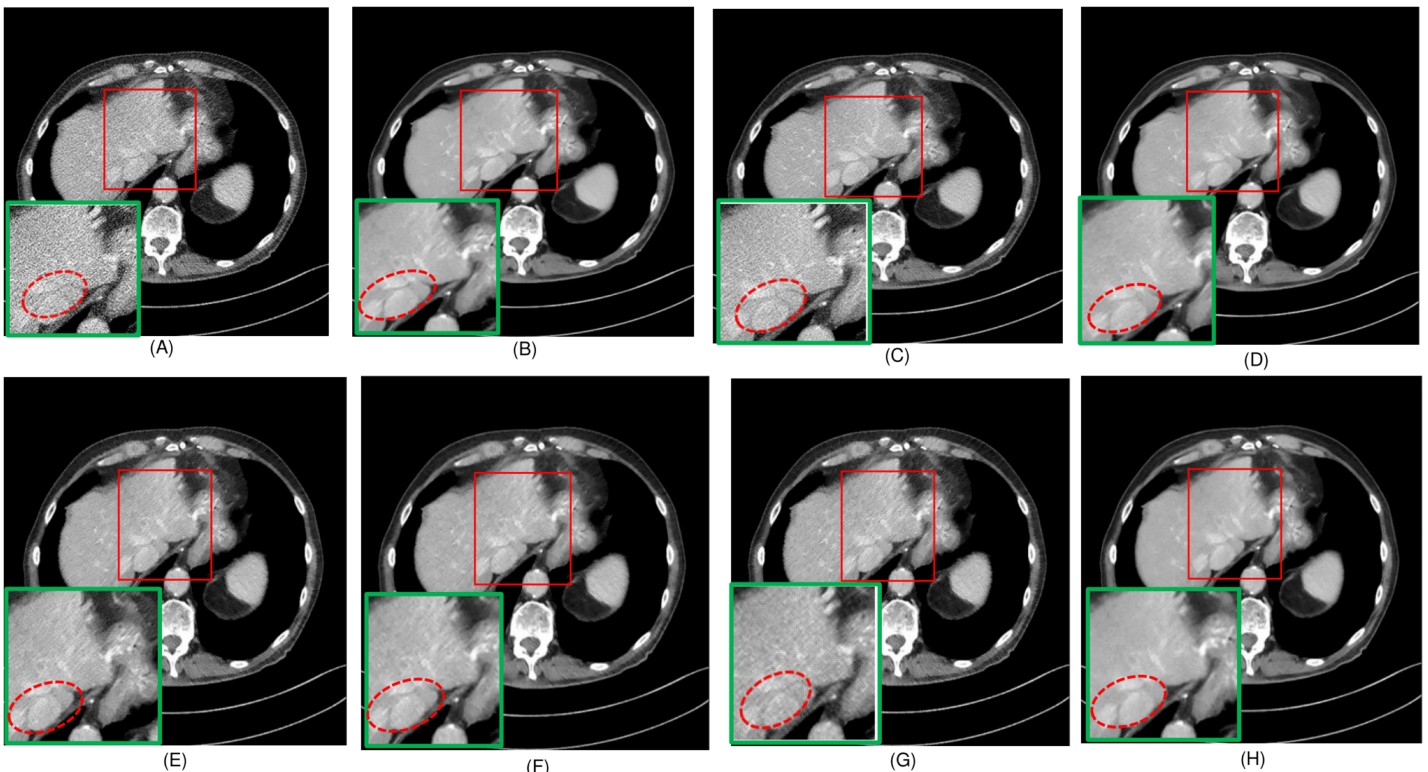

**Figure 6** Overall comparison of test sample 2 (A) LDCT (B) NDCT (C) Uformer (D) Restormer (E) EDCNN (F) CTformer (G) DDT (H) Proposed.

image and demonstrates enhanced preservation of local details within ROI compared to its counterparts.

# ABLATION EXPERIMENTS AND ANALYSIS

## Impact of different network structures on performance

Table 5 compares the effects of different structures within Hybrid Former on enhancing the model's denoising performance. By removing the DFRM from the model and replacing the RSTB in the network structure with an equivalent number of Swin Transformer layers, while keeping other parts unchanged, we obtained a U-shaped network serving as the "Baseline". Subsequently, we replaced the Swin Transformer layers in the Baseline with the CATU to verify its enhancement on model performance. After incorporating CATU, the peak signal-to-noise ratio (PSNR) improved by approximately 0.1, and the structural similarity index measure (SSIM) increased by about 0.001, indicating that introducing the CATU structure effectively boosts the model's performance. Additionally, on top of the Baseline, we employed a parallel structure combining Swin Transformer layers with residual blocks, referred to as "Use RSTB without CCAFL" in the table, to validate whether the integration of CNN and Transformer proposed in this article can significantly enhance model performance. Compared to the Baseline, combining CNN and Transformer yields notable improvements in performance metrics. When experimenting with the complete

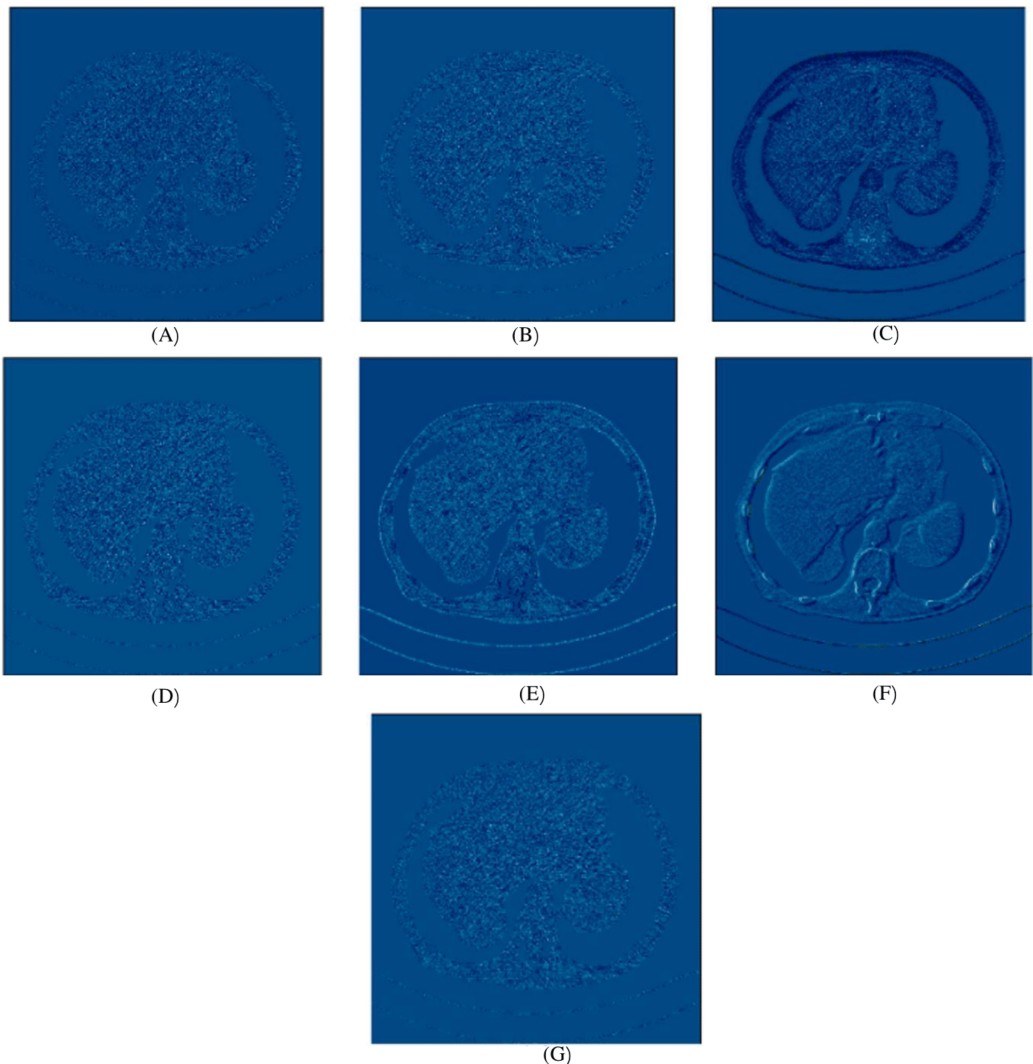

**Figure 7 Noise distribution image of sample 2, after denoising LDCT, and different algorithms (A) LDCT; (B) Restormer; (C) Uformer; (D) CTformer; (E) EDCNN; (F) DDT; (G) Proposed.**

RSTB structure proposed in this article to replace the Swin Transformer layers in the Baseline, denoted as "Use RSTB", both PSNR and SSIM show improvements, proving the effectiveness of the RSTB structure in enhancing the model's performance. Finally, by incorporating the DFRM module, we obtained the complete structure of the proposed model, achieving the best evaluation scores.

### Comparison between CCAFL and channel attention layer

Replacing the CCAFL in CATU with a standard channel attention layer (CAL), where CAL and CCAFL share the same number of convolutional layers, batch normalization (BN) layers, pooling layers, and activation functions with identical parameter settings, the experimental results are shown in Table 6. Here, "None" represents the results without using any channel attention block. It can be observed that the attention layer in CCAFL

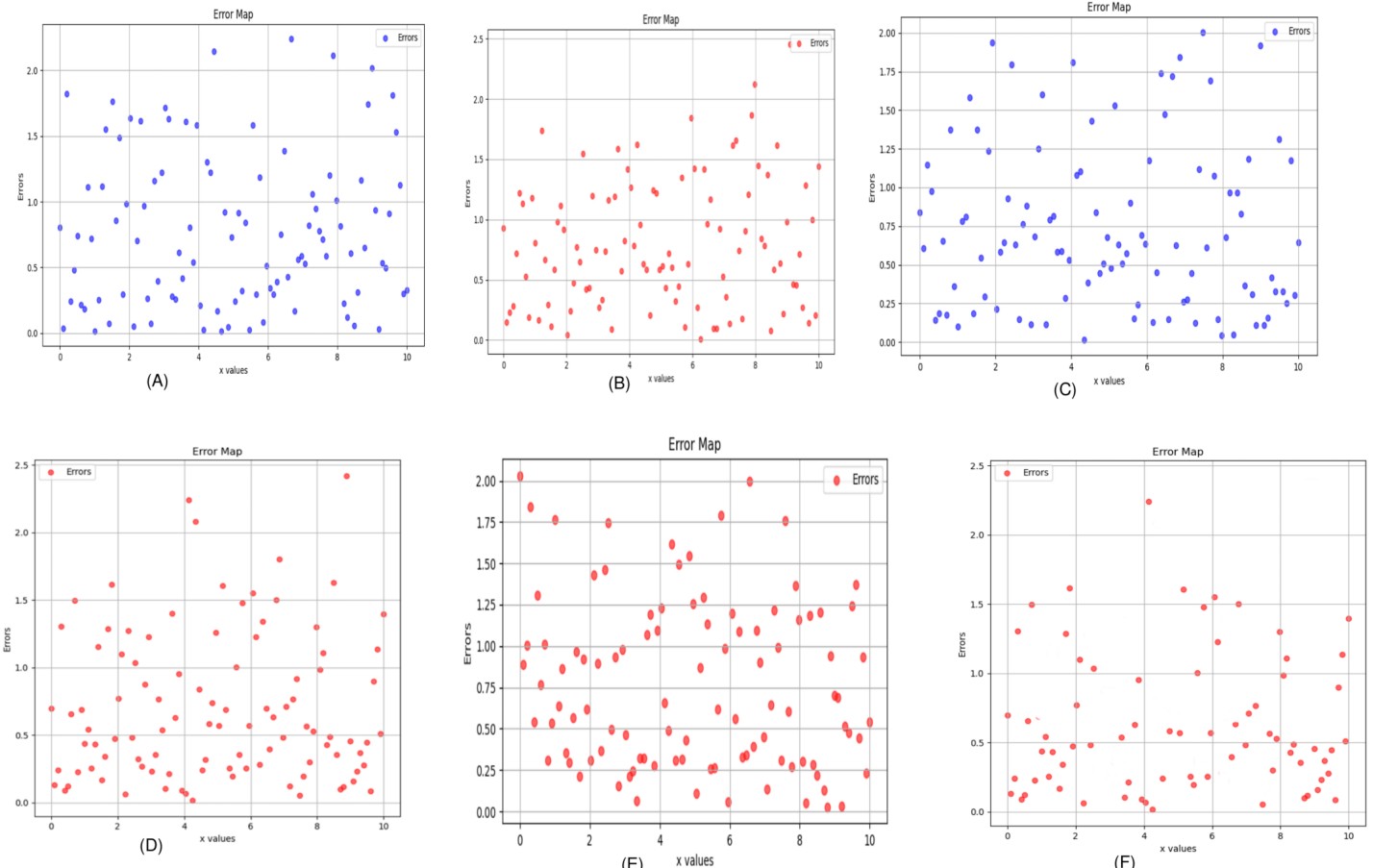

**Figure 8 Error map of the algorithm (A) Uformer (B) Restormer (C) EDCNN (D) CTformer (E) DDT (F) Proposed.**

**Table 2 Evaluation metric results for test sample 1 and test sample 2.**

| Methods | Test sample 1 | | | Test sample 2 | | |
|---|---|---|---|---|---|---|
| | PSNR | SSIM | FSIM | PSNR | SSIM | FSIM |
| LDCT | 18.0145 | 0.7146 | 0.6037 | 19.5628 | 0.7410 | 0.6267 |
| EDCNN (*Liang et al., 2020*) | 24.1536 | 0.7719 | 0.6385 | 26.0578 | 0.8169 | 0.6454 |
| CTformer (*Dayang et al., 2023*) | 23.1605 | 0.7290 | 0.6304 | 24.2325 | 0.8018 | 0.6435 |
| Restormer (*Zamir et al., 2022*) | 24.3402 | 0.7827 | 0.6493 | 26.3010 | 0.8271 | 0.6652 |
| Uformer (*Wang et al., 2022*) | 22.3337 | 0.7253 | 0.6495 | 23.9155 | 0.7971 | 0.6638 |
| DDT (*Liu et al., 2023*) | 21.3093 | 0.7245 | 0.6454 | 20.7690 | 0.7880 | 0.6137 |
| Hybrid former (Proposed) | 24.3610 | 0.7848 | 0.6529 | 26.3256 | 0.8304 | 0.6680 |

receives outputs from all convolutional layers, capturing pixel information from input features to a greater extent and more effectively addressing the issue of texture restoration errors in the Swin Transformer structure.

**Table 3 Evaluation metric results for test sample 3.**

| Methods | PSNR | SSIM | FSIM |
|---|---|---|---|
| LDCT | 18.5628 | 0.7410 | 0.6267 |
| EDCNN (*Liang et al., 2020*) | 22.8693 | 0.7823 | 0.6454 |
| DDT (*Liu et al., 2023*) | 22.2613 | 0.7652 | 0.6435 |
| Restormer (*Zamir et al., 2022*) | 24.3028 | 0.7790 | 0.6552 |
| Uformer (*Wang et al., 2022*) | 24.3155 | 0.7771 | 0.6638 |
| CTformer (*Dayang et al., 2023*) | 24.3600 | 0.7800 | 0.6637 |
| Hybrid Former | 24.3713 | 0.7982 | 0.6705 |

**Table 4 Comparison of experimental results of different algorithms.**

| Methods | PSNR | SSIM | FSIM |
|---|---|---|---|
| LDCT | 21.6048 | 0.8017 | 0.6481 |
| EDCNN (*Liang et al., 2020*) | 25.9639 | 0.8243 | 0.6498 |
| CTformer (*Dayang et al., 2023*) | 26.6431 | 0.8314 | 0.6630 |
| Restormer (*Zamir et al., 2022*) | 27.2664 | 0.8441 | 0.6804 |
| Uformer (*Wang et al., 2022*) | 25.8862 | 0.8393 | 0.6785 |
| DDT (*Liu et al., 2023*) | 24.9213 | 0.8029 | 0.6407 |
| Hybrid Former (Proposed) | 27.3043 | 0.8455 | 0.6835 |

**Table 5 Impact of different structures in the model on denoising performance.**

| Methods | PSNR | SSIM | FSIM |
|---|---|---|---|
| Baseline | 27.1571 | 0.8395 | 0.6808 |
| Use CATU | 27.2588 | 0.8413 | 0.6813 |
| Use RSTB without CCAFL | 27.2761 | 0.8423 | 0.6825 |
| Use RSTB | 27.2890 | 0.8425 | 0.6826 |
| Use RSTB + DFRM (Hybrid Former) | 27.3043 | 0.8455 | 0.6835 |

**Table 6 Impact of different structures in the model on denoising performance.**

| Methods | PSNR | SSIM | FSIM |
|---|---|---|---|
| None | 27.2797 | 0.8427 | 0.6824 |
| With CAL | 27.2843 | 0.8428 | 0.6825 |
| With CCAFL (Hybrid Former) | 27.3043 | 0.8455 | 0.6835 |

**Table 7 Experimental results with different loss functions.**

| Methods | PSNR | SSIM | FSIM |
|---|---|---|---|
| MSE loss | 27.2474 | 0.8416 | 0.6816 |
| Charbonnier loss | 27.2843 | 0.8424 | 0.6824 |
| Compound loss (Hybrid Former) | 27.3043 | 0.8455 | 0.6835 |

**Table 8  Comparison with other Transformer based network.**

| Methods | PSNR | SSIM | FSIM |
|---|---|---|---|
| Swin-UNET | 27.2404 | 0.8436 | 0.6826 |
| Trans-UNET | 27.2953 | 0.8444 | 0.6832 |
| Proposed network (Hybrid Former) | 27.3043 | 0.8455 | 0.6835 |

**Table 9  Presents a comparative analysis of various optimization algorithms.**

| Optimizer | PSNR | SSIM | FSIM |
|---|---|---|---|
| SGD | 24.8835 | 0.8223 | 0.6627 |
| Nadam | 26.6925 | 0.8392 | 0.6781 |
| Adam | 26.8306 | 0.8317 | 0.6628 |
| Adamw | 27.3043 | 0.8455 | 0.6835 |

**Table 10  Showcases the experimental outcomes resulting from the input of images with different sizes.**

| Size | PSNR | SSIM | FSIM |
|---|---|---|---|
| 32 × 32 | 24.2409 | 0.8417 | 0.6816 |
| 64 × 64 | 27.3043 | 0.8455 | 0.6835 |

## Comparison of different loss functions

Table 7 compares the model's denoising performance when using MSE loss, Charbonnier loss, and a composite loss function, respectively. It can be observed that the composite loss function employed in this article achieves more efficient denoising results than using either MSE loss or Charbonnier loss alone.

## Comparison with existing Transformer-based models

As shown in Table 8, the network introduced in this article achieves a PSNR of 0.009%, outperforming numerous prevalent model architectures and establishing the highest denoising metrics to date. This exceptional performance marks a 0.0639 dB (PSNR) improvement over Swin-UNET (*Cao et al., 2022*), which has a comparable infrastructure, and a 0.009 dB (PSNR) enhancement compared to TransUNET (*Chen et al., 2021*). Our method exhibits superior precision in delineating the boundaries of the right ventricle when compared to other transformer-based approaches, further confirming that the proposed method surpasses existing techniques in understanding structural morphology and characteristics.

## Impact of the different optimization algorithms and different image sizes

Tables 9 and 10 present a comparative analysis of various optimization algorithms and image sizes using PSNR, SSIM, and FSIM as performance metrics. SGD serves as a baseline, with Nadam and Adam offering incremental improvements. AdamW, however, stands out with the highest metrics. Additionally, larger image sizes yield better

performance metrics. In conclusion, AdamW and higher-resolution images significantly enhance image quality and similarity assessments.

## CONCLUSION AND FUTURE OUTLOOK

This article introduces a residual convolution and swin transformer fusion network to overcome limitations in extracting global features from CT images using full CNNs and preserve key image details beyond current post-processing methods. While our proposed algorithm (Hybrid Former) framework underwent rigorous validation and evaluation on the widely utilized AAPM dataset and successfully denoises low-dose CT images, there is still ample opportunity for further enhancement and refinement. In future projects, our goal is to explore the extensive capabilities of the Hybrid Former model across a wider range of larger datasets and images with higher resolution. We plan to train and test this data on devices from various manufacturers and apply it to different patients. This will help us gain a deeper understanding of the model's performance and adaptability in real-world scenarios, enabling us to further refine and improve its accuracy and reliability. Additionally, we will strive to enhance the generalization capabilities of the network structure. Building upon our impressive results across various image quality evaluation metrics. In the next stage of our research, we plan to enhance the correlation between denoising tasks and subsequent image processing or analysis tasks to bolster the overall effectiveness and utility of our model. Additionally, we will refine the model to address artifact introduction during the denoising process, ensuring high fidelity and accuracy in the output images. We will meticulously review and improve every aspect of the model and experimental setups to ensure the robustness and reproducibility of our results. Our expected research objectives include developing a high-performance LDCT denoising model that effectively reduces noise and preserves fine details with strong generalization capabilities. Furthermore, we aim to optimize the model to improve its correlation with downstream tasks, enhancing the recovery of important image regions and consequently, the diagnostic value of the denoised images.

### Funding

This study was supported by the Science and Technology Research Project of Higher Education in Hebei Province (Grant No. ZD2022115). The funders had no role in study design, data collection and analysis, decision to publish, or preparation of the manuscript.

### Grant Disclosures

The following grant information was disclosed by the authors:
Science and Technology Research Project of Higher Education in Hebei Province: ZD2022115.

## Competing Interests

The authors declare that they have no competing interests.

## Author Contributions

- Shanaz Sharmin Jui performed the experiments, analyzed the data, performed the computation work, prepared figures and/or tables, and approved the final draft.
- Zhitao Guo conceived and designed the experiments, authored or reviewed drafts of the article, and approved the final draft.
- Rending Jiang analyzed the data, authored or reviewed drafts of the article, and approved the final draft.
- Jiale Liu analyzed the data, authored or reviewed drafts of the article, and approved the final draft.
- Bohua Li analyzed the data, authored or reviewed drafts of the article, and approved the final draft.

## Data Availability

The network training and prediction codes are available at GitHub and Zenodo:

- https://github.com/sharmin93-ctrl/HybridFormer.git
- sharmin93-ctrl. (2025). sharmin93-ctrl/HybridFormer: v0.1.0 (v0.1.0). Zenodo. https://doi.org/10.5281/zenodo.15094096.

The GDP-HMM Challenge datasets, are available at https://www.aapm.org/GrandChallenge/GDP-HMM and https://github.com/RiqiangGao/GDP-HMM_AAPMChallenge/tree/main/data.

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
