# Peer review of "HybridFormer: a convolutional neural network-Transformer architecture for low dose computed tomography image denoising"

_PeerJ Computer Science, doi:10.7717/peerj-cs.2952_

## Round 0.1 · original submission · Major Revisions

Dear Authors,

According to the comments your paper introduce a novel approach that can be of interest for the field. Nevertheless some weakness were detected:
please take into account all the improvements suggested by the Reviewers and implement them.

Best regards,

M.P.

Reviewer 1 ·

Basic reporting

The paper introduces a novel image denoising model leveraging a combination of Residual Convolution and Swin Transformer to address the challenges of noise reduction in low-dose computed tomography (LDCT) imaging. The authors achieve significant quantitative improvements in denoising performance metrics and introduce innovative architectural components like the Combined Attention Transformer Unit (CATU) and a composite loss function. While the manuscript demonstrates novelty and technical rigor, several areas need substantial improvement to enhance its clarity, reproducibility, and scientific depth.

Experimental design

Limited information is provided regarding the data preprocessing steps and the hyperparameter configurations during training. This limits the reproducibility of the results.
The choice of metrics, though standard, would benefit from justification in the context of medical imaging.


The scalability of the proposed architecture for larger datasets or higher-resolution images is not discussed. Potential limitations, such as computational costs or memory constraints, are not adequately addressed.
Figures and Formatting:

Validity of the findings

The paper includes formatting issues, such as inconsistent alignment in tables (e.g., Tables 1–7) and errors in figure captions.
Figures illustrating the architecture (e.g., Hybrid Former structure) are overly complex and lack intuitive explanation.

Critically compare Hybrid Former with closely related models (e.g., Restormer, CTformer) to emphasize its distinct advantages.

Additional comments

Figures illustrating visual comparisons (e.g., Figures 3–6) are difficult to interpret due to poor resolution and inconsistent labeling. Magnified ROI examples require clearer annotations to effectively convey key improvements. Moreover, please move the figures to the main text to improve the readability.

Although the authors cite many recent works, the contextual positioning of Hybrid Former in comparison to the state-of-the-art is insufficient. A deeper analysis of related works (e.g., Restormer, Uformer) is needed to highlight the unique contributions of this work.

Include a discussion on the computational efficiency and scalability of Hybrid Former, particularly in terms of training times, memory usage, and inference speed on real-world data.

Reviewer 2 ·

Basic reporting

no comment.

Experimental design

no comment.

Validity of the findings

no comment.

Additional comments

This paper proposes a hybrid CNN-transformer architecture for LDCT image denoising. The proposed HybridFormer model uses convolution-swin transformer network with CATU and CCAFL units along with RSTB to capture both details and global context. Blurring is avoided via use of perceptual loss in addition to Charbonnier loss. Some decent improvements over prior art is reported. Overall, I find the task examined of great value to the field, and a sound methodology with rather new building blocks for network architecture is proposed by the authors that will be of interest to the readership. At the same time, there are some important issues that should still be clarified to render contributions more transparent.

1) I am not a big fan of reporting PSNR/SSIM etc imporvements in the abstract as in this case they are not that striking. Instead the abstract should try to highlight the cases/results where the proposed model makes a more dramatic impact, could be visual quality, could be improvements over some of the contenders etc.

2) The current literature survey in the introduction section is a very long bucket list of previous approaches in LDCT denoising. However, I found this hard to navigate. My suggestion would be to introduce a Related Work section, or even if this material remains in Introduction, having subsection titles, providing some general categorization among these previous approaches would help the readers go through it more easily.

3) The literature survey on LDCT image denoising and CT imaging in general is lacking some important prior art such as the use of state-space models as a potential alternative to computationally heavy transformer backbones. Discussions of relevant prior art is critical in this regard to higlight technical differences and justify the claimed contributions. Please discuss previous studies as part of the literature survey to this effect.

4) Please discuss why the Charbonnier loss is used. If smoothing is a concern, would it be possible to adopt adversarial of diffusion type losses instead?

5) Since onely slices from 1 patient are used for testing, either cross-validation with different splits should be reported, or the authors should include a clear limitation statement regarding this experimental design choice.

6) In visual displays, it is difficult to see differences between models, for instance in Fig. 3. Please include some error maps, and show red regions at higher zoom level or similar to better illustrate your points, rather than presenting them separately in different figures. I would recommend including more examples as well.

7) In some cases, quantitative benefit over Restormer are limited. This point should be discussed, and an evaluation of whether the observed differences are statistically significant could also be conducted.

8) A discussion on technical and study limitations would be useful.

Reviewer 3 ·

Basic reporting

Summary:
The paper presents a novel image denoising model called HybridFormer for Low-Dose CT (LDCT) images. The model combines Convolutional Neural Networks (CNNs) and the Swin Transformer architecture to enhance the quality of denoising while preserving critical details. The proposed method integrates residual convolutions for local feature extraction and the Swin Transformer for global feature extraction. Additionally, the introduction of the Combined Attention Transformer Unit (CATU) and the Cross-Channel Attentive Fusion Layer (CCAFL) are promising advancements. The paper shows significant improvements in performance on the Mayo dataset, outpacing several existing state-of-the-art methods in terms of PSNR, SSIM, and FSIM.

Weakness:
1、The model might be computationally intensive due to its hybrid architecture. A discussion on its runtime, training time, and hardware requirements would be useful for practical deployment.
2 In some places, terminology can be slightly more consistent. For example, HybridFormer and Hybrid Former should be used consistently throughout the paper.

Please consider to comment more works published recently

Experimental design

The hybrid approach of combining CNNs and Transformers is well-executed, but the novelty of the method in comparison to other Transformer-based denoising models is not sufficiently highlighted. A clear comparison with existing models (such as SwinUNET, ViT-UNet, or TransUNet) is needed to clarify how HybridFormer adds value beyond existing approaches.

One potential limitation of this study is the reliance on a relatively small dataset, the Mayo dataset, for testing the proposed HybridFormer model. While the results are promising, the generalizability of the model to real-world scenarios, especially in clinical settings, may be questioned if the dataset is not diverse enough. A larger and more varied dataset, representing a broader range of patient demographics, CT scanner types, and clinical conditions, would provide a more robust evaluation of the model's performance.

Validity of the findings

1.The combination of CNN and Transformer models is innovative and potentially powerful for image denoising.
2.The authors have clearly articulated the problem (LDCT image denoising) and the motivation behind using a hybrid architecture.
3.The HybridFormer model demonstrated substantial improvements over existing state-of-the-art methods in quantitative metrics (PSNR, SSIM, FSIM).
4.The proposed Deep Feature Refinement Module (DFRM) and loss function combination show promise in preserving image details, which is critical in medical imaging.

Additional comments

Please consider cite [1,2]

Reviewer 4 ·

Basic reporting

It is recommended to further explain the HybridFormer in abstract and how it exactly work?
Authors suppose to include why are the health issues come due to X-Rays and what the limitations of the existing literature/systems by not efficiently working well for computer?

Introduction part is long and required to be concise in wording selection. highlight the contributions work.

Experimental design

It is appreciated to share source code of this simulation experiments via Github.com,etc.

It is required to mention the dataset used in this simulation.

Validity of the findings

Results need further improvements why the proposed work is better than the existing work.
Results should be mentioned in the accumulative percentage for each performance parameters.

---

## Round 0.2 · accepted · Accept

Dear Authors,

the comments raised by the reviewers were addressed.

One reviewer pointed out that maybe some important references should be discussed such as 10.1109/TGRS.2024.3446812, 10.1109/TGRS.2024.3374953, considering their high relevants.

I would like to remind that it is PeerJ policy that additional references suggested during the peer-review process should only be included if the authors are in agreement that they are relevant and useful.

Given that, adding those references it is up to you.

Best,

M.P.

Reviewer 1 ·

Basic reporting

This paper uses residual convolution for local feature extraction and Swin Transformer for global feature extraction, boosting denoising efficacy.

Experimental design

no comment

Validity of the findings

no comment

Additional comments

Some important references should be discussed such as 10.1109/TGRS.2024.3446812, 10.1109/TGRS.2024.3374953, considering their high relevants.

Reviewer 4 ·

Basic reporting

All my comments addressed and accept it.

Experimental design

Well, experimental work has been properly prepared and presented which is sufficient at this stage.

Validity of the findings

The proposed work has validated via different ways and accepted.

Additional comments

no comments, accept